# MINIM: Privacy-Aware Minimal View for Agents via Trusted Local Sanitization

**Hexuan Yu** [1]  **Chaoyu Zhang** [1]  **Heng Jin** [1]  **Shanghao Shi** [2]  **Ning Zhang** [2]  **Y. Thomas Hou** [1]  **Wenjing Lou** [1]

## Abstract

Modern LLM-powered autonomous agents increasingly rely on rich user interface (UI) state observations to achieve reliable action grounding in complex digital environments. However, many deployments transmit the full UI state to remote inference servers even when most elements are irrelevant to the current task, which can leak sensitive but unnecessary context such as authentication codes, private notifications, and background application states. We propose MINIM, a trusted local broker that performs privacy-aware minimization on the client side before any observation leaves the device. Grounded in Contextual Integrity (CI), MINIM learns a dual-score representation for each UI element by predicting an inherent sensitivity score ($s$) and a task-conditioned necessity score ($n$). These scores drive a ternary disclosure policy that keeps essential elements, abstracts sensitive attributes when needed, and removes task-irrelevant content. We optimize a CI-aware objective that penalizes necessity errors more strongly on high-risk content, enabling aggressive pruning while preserving task-critical information. Experiments on real-world UI observations derived from WebArena show that MINIM substantially reduces task-irrelevant sensitive leakage while preserving task-critical semantic context and the interactive affordances required for reliable agent actions.

## 1. Introduction

Modern agentic systems increasingly interact with the digital world through *structured observations*, which represent interface state using explicit semantic structure rather than raw sensory streams. Prior work on interface-grounded agents has explored both pixel-based inputs for web and GUI reasoning (He et al., 2024; Koh et al., 2024) and structured hierarchies such as accessibility trees or DOM-like representations (Deng et al., 2023; Zhou et al., 2024). A prominent instantiation is the *accessibility API*, which exposes a hierarchy of UI elements with roles, states, and affordances, and is widely used in OS-level assistants (e.g., Apple Intelligence (Apple., 2024) and Microsoft Copilot (Microsoft, 2024)) due to its stability relative to pixel-based inputs (Nguyen et al., 2025). More broadly, structured observations also appear as DOM representations in web agents, scene graphs in robotics, and tool-use schemas (e.g., the Model Context Protocol (MCP) (Anthropic, 2024)) that standardize tool definitions and invocation via typed schemas.

However, these interfaces were primarily designed for assistive transparency rather than privacy-aware orchestration. As a result, many agent deployments follow a share-first design, sending rich interface state to remote inference to simplify integration and latency engineering. In our primary setting, this means disclosing the *entire* accessibility tree even when only a small subset is required for the user task. We term this failure mode *Semantic Over-Privileged Observation*, where task-irrelevant elements in a structured hierarchy are exposed together with their functional semantics. For example, during a routine request such as "Summarize this email", the remote agent may observe co-located UI context such as sidebar notifications, background application windows, or unrelated browser tabs. This can leak personally identifiable information and cross-session behavioral traces that are irrelevant to the immediate task but useful for profiling. Such risks have been quantified for autonomous agents and demonstrated in recent attacks (Zharmagambetov et al., 2025; Liu et al., 2025; Shao et al., 2024; Carlini et al., 2021; Green et al., 2025).

Safeguarding structured observations for *autonomous agents* is challenging because what should be disclosed is inherently *task-conditioned*. The same element may be necessary for completing one task (e.g., a 2FA code needed to authenticate) yet constitute pure leakage under another intent (e.g., browsing or summarization). Recent work further highlights a "Privacy Judgment-Action Gap" (Wang et al., 2025b), where agents fail to protect context even when it is recognized as sensitive. Existing privacy paradigms struggle

[1]Virginia Tech, Arlington and Blacksburg, VA, USA [2]Washington University in St. Louis, St. Louis, MO, USA. Correspondence to: Hexuan Yu <hexuanyu@vt.edu>.

*Proceedings of the 43rd International Conference on Machine Learning*, Seoul, South Korea. PMLR 306, 2026. Copyright 2026 by the author(s).

to address this setting. *Task-agnostic entity filtering* (e.g., Presidio (Microsoft, 2024)) relies on static PII categories, which can remove task-critical context or miss sensitive non-PII attributes (e.g., political preference) (Kim et al., 2024; Garza et al., 2025). *Differential Privacy (DP)* introduces stochastic perturbations that can distort the precise semantic cues required for reliable actuation in structured interfaces (Zhang et al., 2025a; Abadi et al., 2016; Yu et al., 2021). Finally, *cryptographic LLM methods* (Pang et al., 2024; Riasi et al., 2025; Xu et al., 2025; Rathee et al., 2020) (e.g., Multi-Party Computation (MPC), Fully Homomorphic Encryption (FHE)) safeguard computation but do not prevent sensitive inference from whatever state is disclosed to the remote server, and they often incur latency that is incompatible with real-time agentic control loops. Distinct from conversational safeguards for user prompts (Ngong et al., 2025; Zhou et al., 2025), meaningful privacy for autonomous agents requires minimizing the *structured observation* before it is disclosed to the inference server.

To address this, we propose MINIM, a *trusted local broker* that enforces pre-disclosure minimization on the client device. MINIM implements a learned structural bottleneck that sanitizes the agent's observation before it is transmitted to a remote inference server. Unlike prompt-based sanitizers that rely on the agent's own reasoning, MINIM uses a specialized local model to predict two scalar scores for each element in the structured observation: (1) *sensitivity*, which captures inherent information risk, and (2) *task-conditioned necessity*, which captures utility for the current intent. These scores drive a disclosure policy grounded in *Contextual Integrity* (CI) (Nissenbaum, 2004), restricting disclosure of high-sensitivity content unless it is predicted to be necessary for completing the user's task.

**Contributions.** We make three contributions. First, we identify *Semantic Over-Privileged Observation* as a privacy risk in agentic systems that rely on structured observations, and formalize pre-disclosure minimization as learning a CI-compliant structural bottleneck. Second, we propose MINIM, which decouples contextual scoring from policy enforcement via a normative policy layer driven by joint sensitivity and necessity prediction, trained with a CI-aware objective that penalizes unnecessary disclosure of high-risk content. Finally, we evaluate MINIM on real agent observations instantiated as accessibility trees derived from WebArena (Zhou et al., 2024) across multiple domains (Shopping, Reddit, and Gmail), demonstrating substantial reduction in task-irrelevant sensitive leakage while preserving task-critical content.

## 2. Problem Setup and Preliminaries

We focus on privacy-preserving perception for autonomous agents that act upon structured observations. While our

framework is generalizable to various hierarchical state descriptions, we instantiate and evaluate our method on accessibility trees, which serve as the primary interface for modern OS-level agents.

**Structured Interface Representation.** Let $X_t$ denote the raw structured observation at time $t$, modeled as a hierarchical tree of elements $\{e_i\}_{i=1}^N$. Each element $e_i$ is characterized by a set of attributes including its semantic role (e.g., button, heading), text content, interaction state (e.g., checked, disabled), and structural properties such as depth and lineage. Unlike pixel-based inputs, this representation provides a discrete and semantically rich basis for agent reasoning.

**Agent Interaction Model.** The agent operates in a sequential observation-action loop conditioned on a user task $T$. At each time step $t$, the agent samples an action $a_t$ from its policy $a_t \sim \pi(\cdot \mid Z_t, T)$, where $Z_t$ represents the observation exposed to the remote inference server. In standard deployments, the server receives the full raw state ($Z_t = X_t$). Our goal is to interpose a local transformation function $f$ to produce a sanitized view $Z_t = f(X_t, T)$ which minimizes sensitive information leakage while preserving the utility required to complete $T$. Crucially, $Z_t$ is not strictly a subset of $X_t$, as $f$ may involve both the pruning of elements and the abstraction of specific attributes.

**Contextual Integrity.** CI posits that privacy is governed by adherence to appropriate information flow norms rather than absolute secrecy (Nissenbaum, 2004). These norms are characterized by four parameters: *contexts*, *actors* (senders and receivers), *attributes* (types of information), and *transmission principles* (constraints on how information may flow). Recent work has begun to adopt CI as a normative lens for disclosure and social reasoning in LLM-based assistants and dialogue agents (Lan et al., 2025; Mireshghallah et al., 2023; Tan et al., 2026). Our work extends this perspective to the observation channel of agents by operationalizing task-conditioned necessity over structured elements and their attributes prior to disclosure.

In our setting, the user's active task $T$ establishes the *context*, while the client device and remote inference server serve as the *actors*. The data fields within $X_t$ correspond to the *attributes*. We target a *transmission principle* of task-conditioned necessity, which requires that information with high disclosure risk be shared only when it is essential for completing the current task. We operationalize this principle by learning predictive signals for sensitivity ($s_i$) and necessity ($n_{i,T}$) to drive the disclosure policy.

**Threat Model.** We assume MINIM is deployed locally as a trusted broker that intercepts the agent's raw observation

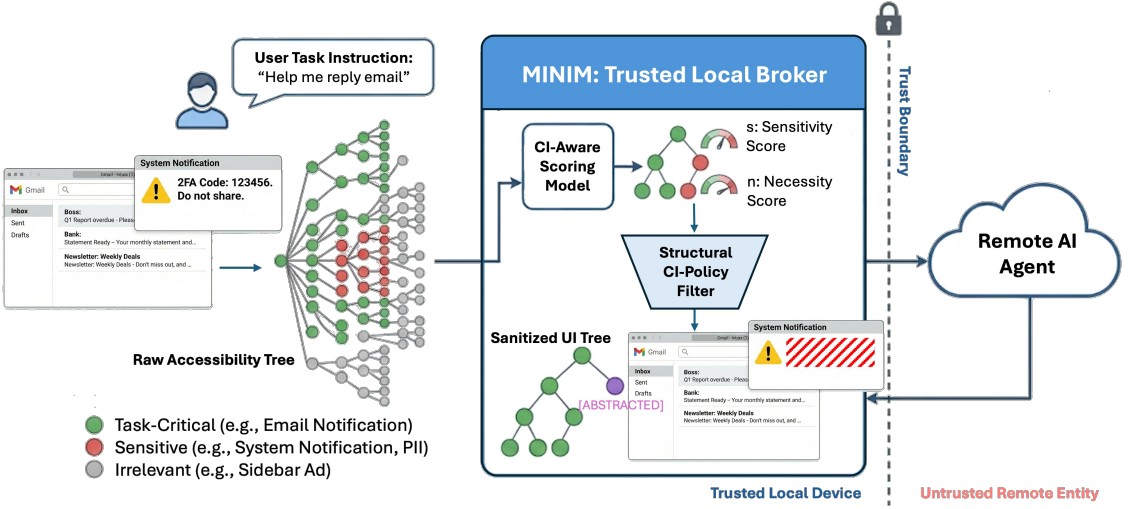

*Figure 1.* System Architecture of MINIM. The trusted local broker intercepts raw structured observations (e.g., accessibility trees) and performs contextual integrity-driven sanitization. By jointly predicting sensitivity and task-conditioned necessity, MINIM implements a structural bottleneck that filters or abstracts information before it is transmitted to the remote agent.

and outputs a sanitized view before any data is transmitted to a remote inference server. Our threat model excludes compromise of the local environment that hosts MINIM, including OS-level malware and privileged attackers. The remote agent is honest-but-curious: it executes the task according to the protocol given the disclosed observation and task description, while passively harvesting task-irrelevant content (e.g., background windows) for downstream profiling or inference. We do not consider active adversaries that attempt to manipulate the agent via prompt injection or malicious content, as such attacks are orthogonal to our focus on pre-disclosure observation minimization.

## 3. Related Work

Prior defenses for LLM systems largely focus on unstructured language, identifying sensitive content through pattern-based detectors or user intervention. User-centric tools such as PrivWeb (Zhang et al., 2025b) adopt human-in-the-loop filtering, which does not scale well to autonomous workflows. Automated approaches, including local gateways and intermediaries (e.g., AirGapAgent (Bagdasarian et al., 2024), Portcullis (Zhan et al., 2025), and Papillon (Li et al., 2025)) and PII redaction benchmarks and evaluations (Shen et al., 2025; Sun et al., 2025), primarily target conversational prompts, logs, or free-form text. In parallel, recent work has begun to quantify privacy leakage and data minimization objectives in agentic settings (Zharmagambetov et al., 2025; Wang et al., 2025b). While these approaches provide

important baselines for reducing disclosure, they do not directly address agents whose observations are *structured* and action-critical. In such settings, naive redaction or perturbation can break the structural and semantic cues needed for reliable decision making and actuation. Our work focuses on pre-disclosure minimization for structured observations under task-conditioned norms, and we instantiate and evaluate the approach on accessibility trees. Complementary work systematizes broader agent security and privacy threat surfaces (Yu et al., 2025; He et al., 2025), analyzes privacy risks from agent memory and logging (Wang et al., 2025a; Liu et al., 2026).

## 4. Methodology

### 4.1. Overview

We study pre-disclosure minimization for agent observations under CI. The core tension is that a remote agent needs some UI context to act reliably, but sending the full structured state (e.g., an accessibility tree) often reveals sensitive, task-irrelevant information. Our goal is to minimize sensitive disclosure while still transmitting the minimum necessary information required by the current task context.

Figure 1 illustrates this process through a concrete example. A user shares a screenshot of their desktop with a remote agentic AI system to help reply to an email. The raw interface state includes both the email content and unrelated UI context, such as a system notification displaying a verifica-

tion code. Although this code is sensitive, it is irrelevant to the email-reply task. Under our scheme, a trusted local broker intercepts the raw observation before it is sent to the remote agent. Conditioned on the task, the broker assigns each UI element an inherent sensitivity score and a task-conditioned necessity score. A well-trained model identifies the verification code as **highly sensitive but unnecessary** and abstracts it (replacing the raw value with a placeholder while preserving its structural role), while retaining the email text and other interface elements required for drafting a reply. The resulting sanitized view, which preserves task-critical context while suppressing irrelevant sensitive information, is the only observation transmitted to the remote agent.

### 4.2. Training for Context-Conditioned Scoring

Our training phase consists of two stages: constructing a training dataset that captures task-conditioned privacy norms, and learning a scoring model whose objective operationalizes CI.

We build a data pipeline that produces labeled accessibility trees across domains and tasks (e.g., replying to an email while an unrelated system notification displays a verification code). ① We first collect the raw accessibility tree $X_t = \{e_i\}_{i=1}^N$ from the current UI state, where each node $e_i$ contains its text attributes, accessibility role, and structural metadata. ② We then pair $X_t$ with the user task $T$, which defines the active context of the interaction. ③ Given the scene and task, we assign each node a task-conditioned necessity score $n_{i,T}$, measuring how essential the node is for completing $T$, with the highest scores given to elements directly required to execute the user intent. ④ In parallel, we annotate each node with a task-independent sensitivity score $s_i$, capturing the inherent disclosure risk of the information it carries, and explicitly identifying content that is sensitive yet unnecessary under the current task. This process yields fully labeled accessibility trees in which every node is associated with a score pair $(s_i, n_{i,T})$. Using these labeled trees, we can train a model to act as a scoring module. Given node-level features and the task description, the model predicts a pair of scores $(\hat{s}_i, \hat{n}_{i,T})$ for each node.

The training objective is designed to approximate the CI goal of minimizing inappropriate disclosure while preserving task success. Formally, we aim to minimize the disclosure of contextually inappropriate information subject to a task utility constraint:

$$\min_{Z_t} \ I(X_t^{\text{out}}; Z_t \mid T) \quad \text{s.t.} \quad \text{TaskSuccess}(Z_t, T) \geq \tau,$$
(1)

where $X_t^{\text{out}}$ denotes flows that violate CI norms. Because $X_t^{\text{out}}$ is latent, we optimize this objective through the two learned proxies, sensitivity $s$ and necessity $n$.

We supervise sensitivity prediction using an absolute error loss:

$$\mathcal{L}_{\text{sens}} = \sum_i |s_i - \hat{s}_i|,$$
(2)

where $s_i$ denotes the ground-truth sensitivity of node $e_i$ and $\hat{s}_i$ is the model's prediction. This term encourages accurate estimation of inherent disclosure risk independent of task context.

To encode CI weights, we define a necessity loss that is weighted by sensitivity risk:

$$\mathcal{L}_{\text{nec}} = \sum_i \left(1 + \lambda \cdot \frac{s_i}{10} \cdot \left(1 - \frac{n_{i,T}}{10}\right)\right) \cdot |n_{i,T} - \hat{n}_{i,T}|, \ (3)$$

where $n_{i,T}$ is the ground-truth necessity of node $e_i$ under task $T$, $\hat{n}_{i,T}$ is the predicted necessity, and $\lambda$ controls the strength of CI weighting. The multiplicative factor increases the loss when a node is highly sensitive ($s_i$ large) yet task-irrelevant ($n_{i,T}$ small), which is the regime where inappropriate disclosure is most harmful. The final objective combines both terms: $\mathcal{L} = \mathcal{L}_{\text{sens}} + \alpha \mathcal{L}_{\text{nec}}$, where $\alpha$ balances sensitivity accuracy and task-conditioned utility. By penalizing necessity errors more strongly on high-risk, low-utility nodes, this objective aligns optimization with the goal of suppressing task-irrelevant sensitive information while preserving context required for task completion.

### 4.3. Task-Conditioned Disclosure at Deployment

We now describe how a trained model is used during deployment. At runtime, a user issues a task $T$ (e.g., replying to an email), which triggers an agentic workflow on a concrete application. As part of this workflow, the application exposes a structured observation in the form of an accessibility tree $X_t = \{e_i\}_{i=1}^N$. Before this observation is transmitted to a remote agentic AI system, MINIM operates as a local broker that mediates disclosure.

Given the task $T$ and the observed accessibility tree $X_t$, the trained scoring model is applied to each node to predict a pair of scores $(\hat{s}_i, \hat{n}_{i,T})$, where $\hat{s}_i$ estimates the inherent sensitivity of node $e_i$ and $\hat{n}_{i,T}$ estimates its necessity for completing task $T$. These scores are then consumed by a fixed decision procedure that enforces task-conditioned minimization.

Specifically, each node is mapped to one of three actions—R, A, or K —based on thresholded comparisons:

$$\text{Action}(e_i) = \begin{cases} \mathsf{R} & \text{if } \hat{n}_{i,T} < \tau_{\text{nec}}, \\ \mathsf{A} & \text{if } \hat{n}_{i,T} \geq \tau_{\text{nec}} \ \wedge \ \hat{s}_i \geq \tau_{\text{sens}}, \\ \mathsf{K} & \text{if } \hat{n}_{i,T} \geq \tau_{\text{nec}} \ \wedge \ \hat{s}_i < \tau_{\text{sens}}. \end{cases} \tag{4}$$

Nodes predicted as unnecessary are removed (R). Nodes that are necessary but sensitive are abstracted (A), meaning

that sensitive attributes are replaced with placeholders while preserving structural roles. Nodes that are both necessary and low-risk are kept unchanged (K).

Applying this procedure to all nodes produces a sanitized accessibility tree $Z_t$ in which disclosure is gated by predicted necessity and attribute fidelity is modulated by predicted sensitivity, jointly enforcing CI at deployment. Algorithm 1 summarizes the full procedure.

# 5. Experiments

## 5.1. Experimental Setting

Our experiments evaluate whether MINIM achieves its intended design goals: reducing task-irrelevant privacy leakage, preserving actionable utility required for task completion, and generalizing across domains and task semantics.

We construct a privacy-augmented corpus from WebArena (Zhou et al., 2024) spanning *Shopping*, *Reddit*, and *Gmail*. The corpus contains 150 unique accessibility trees paired with 27 task types, yielding 5,403 (tree, task) variants. We inject synthesized sensitive context (e.g., 2FA codes, password prompts, system notifications) during preprocessing. Each UI element receives a task-independent sensitivity score $s_i \in [0, 10]$ and a task-conditioned necessity score $n_{i,T} \in [0, 10]$. The dataset is split into 4,741 training and 662 test variants (Appendix C). We report all results on the held-out test set ($N = 662$). We evaluate MINIM using a scoring model trained with the CI-aware objective (Appendix B). As this work is among the first to study task-conditioned, pre-disclosure minimization for structured agent observations, we design comparisons within this setting to isolate the effects of our design choices. Specifically, we consider two baseline families: (i) Fixed Policies, which apply alternative disclosure rules given the same MINIM-predicted scores (*Full Observation*, *Sensitivity-Only*, *Necessity-Only*, *Random Budget*); and (ii) Prompted LLM Scorers, where open-weight LLMs are prompted to output $(s, n)$ scores using our annotation rubric (Appendix E).

Sanitization rarely eliminates privacy risk entirely (Shen et al., 2025; Ngong et al., 2025; Garza et al., 2025); we therefore model A actions as incurring non-zero residual leakage. For a node $i$ with action $a_i \in \{K, A, R\}$, we define its post-sanitization token contribution $\text{tok}_{\text{post}}(i)$ as $\text{tok}(i)$ if $a_i = K$, $\max(\text{tok}_{\text{ph}}, p_{\text{abs}} \cdot \text{tok}(i))$ if $a_i = A$, and 0 if $a_i = R$. We set $\text{tok}_{\text{ph}} = 6.0$ and $p_{\text{abs}} = 0.05$ for the primary analysis, and report sensitivity to this choice up to $p_{\text{abs}} = 0.1$.

## 5.2. Evaluation Metrics

**TCNP (Task-Critical Node Preservation).** TCNP measures the fraction of task-critical elements that remain visible after sanitization, capturing preservation of semantic context. For example, in an email-reply task, TCNP reflects whether the email body and relevant headers are retained. Letting $\mathcal{C}(X_t, T) = \{i \mid n_{i,T} \geq \tau_{\text{crit}}\}$ denote task-critical elements, we define $\text{TCNP}(X_t, T) = |\mathcal{C}(X_t, T) \cap Z_t|/|\mathcal{C}(X_t, T)|$. Higher TCNP indicates better retention of task-relevant context.

**TCNP-I (Task-Critical Node Preservation-Interactive).** TCNP-I measures whether the sanitized observation preserves the interactive elements required to execute the task, capturing execution viability rather than descriptive completeness. For example, during checkout, TCNP-I reflects whether the *Purchase* button is preserved, regardless of surrounding text. Formally, $\text{TCNP-I}(X_t, T) = \frac{|\{i \in Z_t | n_{i,T} \geq \tau_{\text{crit}} \wedge \text{interactive}(i)\}|}{|\{i \in X_t | n_{i,T} \geq \tau_{\text{crit}} \wedge \text{interactive}(i)\}|}$.

**TISL (Task-Irrelevant Sensitive Leakage).** TISL measures how much sensitive information is disclosed when it is not required for the task. It captures privacy risk by weighting exposed task-irrelevant elements by their sensitivity. For example, revealing a 2FA code during an unrelated browsing task increases TISL. We compute per-instance leakage as $\text{TISL}(Z_t; X_t, T) = \sum_{i \in Z_t} \mathbb{1}[n_{i,T} \leq \tau_{\text{irr}}] \cdot s_i \cdot \text{tok}_{\text{post}}(i)$, and report a dataset-normalized version relative to full observation to ensure comparability across methods.

## 5.3. Main Results

**Utility vs. Privacy Trade-offs.** Table 1 demonstrates that MINIM achieves a highly efficient operating point: TCNP-I 0.9931, TCNP 0.9491, and TISL 0.101 (10.1% of Full Observation). This efficiency reflects the intrinsic sparsity of accessibility trees: empirical studies indicate that modern web pages are increasingly element-heavy, with an average of over 1,100 HTML elements per page (WebAIM, 2024), the vast majority of which serve as structural wrappers (nested "containers"), decorative items, or redundant links rather than actionable affordances. By training explicitly to distinguish these sparse *actionable affordances* (buttons, inputs) from *passive context*, MINIM can aggressively prune the latter without impeding the agent's ability to act.

MINIM's adaptive policy dominates single-score baselines by resolving the utility–privacy tension in Table 1 (TISL is normalized to *Full Observation*). **Sensitivity-Only** yields TCNP 0.0401 and TISL 0.3799: retaining the 20% least-sensitive nodes produces predominantly task-irrelevant content, while task-critical nodes with any sensitivity are discarded. Conversely, **Necessity-Only** achieves TCNP 0.9445

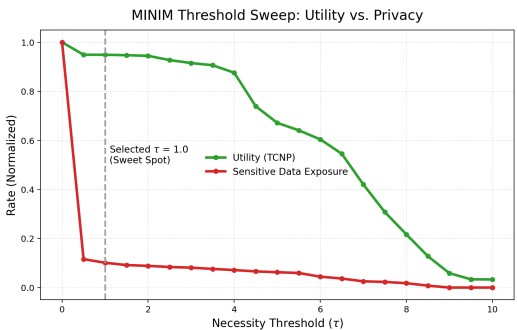

Figure 2. Privacy–utility trade-off under necessity threshold sweep. TCNP (utility) and TISL (leakage) are plotted as a function of $\tau_{\text{nec}}$ swept over $[0, 10]$ in steps of 0.5. The dashed vertical line marks the selected operating point $\tau_{\text{nec}} = 1.0$, which achieves TCNP $\approx 0.949$ and TISL $\approx 0.101$. Increasing $\tau_{\text{nec}}$ reduces leakage monotonically but degrades task utility rapidly beyond $\tau_{\text{nec}} \gtrsim 3$.

Table 1. **Main Results on Augmented WebArena** ($N = 662$). Baselines apply a top-$K$ budget-matching policy retaining 20% of nodes; MINIM uses its adaptive K/A/R policy with default thresholds. (TCNP: Context Recall; TCNP-I: Actionable Recall.)

| Method | TCNP ↑ | TCNP-I ↑ | TISL ↓ |
|---|---|---|---|
| Full Obs. | 1.0000 | 1.0000 | 1.0000 |
| Random Budget | 0.2284 | 0.2346 | 0.1971 |
| Sens.-Only | 0.0401 | 0.0393 | 0.3799 |
| Nec.-Only | 0.9445 | 0.9730 | 0.2032 |
| **MINIM (Ours)** | **0.9491** | **0.9931** | **0.1010** |

but incurs TISL 0.2032 by disclosing sensitive content whenever it is predicted to be useful. **Random Budget** fails both objectives, confirming that effective minimization requires semantic understanding rather than uniform pruning. By jointly modeling both dimensions, MINIM matches or exceeds Necessity-Only in utility (TCNP 0.9491, TCNP-I 0.9931) while halving TISL (0.1010 vs. 0.2032). This adaptive behavior, applying A to high-necessity, high-sensitivity conflict nodes and R to privacy risks and clutter (Figure 3), explains its superior trade-off.

We select the necessity threshold $\tau_{\text{nec}}$ via a validation sweep, balancing utility and leakage: $\tau_{\text{nec}} = \arg\max_\tau(\text{TCNP}(\tau) - 0.5 \cdot \text{TISL}(\tau))$. Inference uses the selected $\tau_{\text{nec}} = 1.0$ (Figure 2). Necessity annotations take values in $\{0, 5, 10\}$; we use $\tau_{\text{crit}} = 7$, $\tau_{\text{irr}} = 1$. Sensitivity threshold is fixed at $\tau_{\text{sens}} = 5.0$.

### 5.4. Comparison with Prompted LLM Scorer Baselines

**Configuration.** We evaluate a representative set of open-weight LLMs as prompted scorers under realistic deployment constraints for privacy-preserving agent pipelines (Table 2). We restrict this comparison to open-source models to reflect the deployability requirements of privacy-sensitive settings, where downstream customization, local integration,

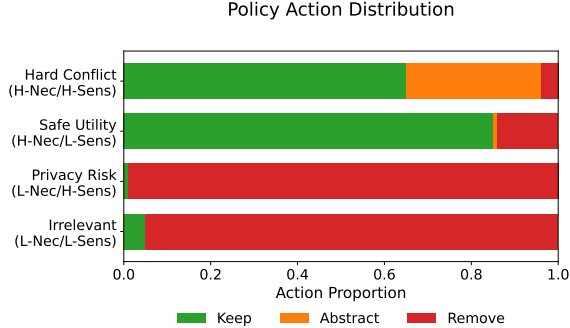

Figure 3. Policy Decision Logic. Distribution of inferred actions (K, A, R) across semantic categories. The model learns to apply A (orange) to nodes in the "Conflict" zone (High Necessity, High Sensitivity) while applying R (red) to privacy risks and clutter. This adaptive behavior explains the performance gap over baseline policies.

Table 2. **Baseline Comparison: LLMs vs MINIM.** Zero-shot privacy performance of leading open-weights LLMs operating on text-only accessibility trees, compared to our specialized approach. TCNP denotes Context Recall; TCNP-I denotes Actionable Recall.

| Method | Context Recall (TCNP) ↑ | TCNP-I ↑ | TISL ↓ | Keep% ↓ |
|---|---|---|---|---|
| Qwen3-8B-Instruct | 0.956 | 0.989 | 0.211 | 28.4 |
| Nemotron-Nano-9B | 0.958 | 0.991 | 0.249 | 29.1 |
| GPT-OSS-20B | 0.963 | 0.994 | 0.276 | 32.7 |
| Llama-3.3-70B-Instruct | 0.966 | 0.996 | 0.312 | 34.5 |
| Mistral-7B-v0.3 | 0.953 | 0.987 | 0.257 | 30.2 |
| Llama-3-8B-Instruct | 0.955 | 0.988 | 0.206 | 27.6 |
| Gemma-3N-E4B | 0.951 | 0.985 | 0.194 | 25.8 |
| MINIM (Ours) | 0.9491 | 0.9931 | 0.101 | 12.0 |

or system-level enforcement is necessary and closed-source models are impractical. We prioritize models with moderate parameter counts, as local deployment is most feasible for lightweight architectures suited to on-device or edge inference. We additionally include Llama-3.3-70B-Instruct as a large-scale reference to assess whether scaling confers measurable benefits, and find only marginal gains despite an order-of-magnitude increase in model size.

Table 2 shows that prompted LLM scorers achieve TCNP-I 0.985–0.996 and TCNP 0.951–0.966, consistent with the broad affordance and content recognition capacity of general instruction-tuned models. However, privacy behavior varies substantially across models: TISL spans 0.194–0.312, substantially above MINIM's 0.101. This gap reveals that general instruction tuning does not enforce task-conditioned minimization; these models retain task-irrelevant sensitive content alongside task-useful elements. For example, Llama-3-8B-Instruct retains 27.6% of nodes yet incurs TISL 0.206, whereas MINIM retains only 12.0% of nodes and achieves TISL 0.101 without any LLM inference at test time. Moreover, larger models tend to retain more sensitive content and incur higher leakage (e.g., Llama-3.3-70B-Instruct TISL 0.312 vs. Llama-3-8B-Instruct 0.206), confirming that

model scale alone does not reliably enforce minimization in the absence of explicit contextual integrity constraints.

**Interpreting Contextual Divergence.** We identify a fundamental privacy–efficiency gap between the two paradigms. LLM baselines achieve marginally higher TCNP (0.951–0.966 vs. 0.9491) by operating as passive readers, retaining approximately 26–35% of nodes and thereby incorporating task-irrelevant content that inflates both recall and leakage simultaneously. By contrast, MINIM functions as an active filter, retaining only 12% of nodes while matching or exceeding the TCNP-I of most LLM baselines (0.9931, surpassing five of seven and within 0.003 of the strongest) at substantially lower leakage. For instance, for a transactional task such as "Checkout," MINIM isolates the target interaction element (e.g., the *Purchase* button) while pruning surrounding product descriptions, reviews, and footer links, thereby preserving comparable TCNP-I without exposing unnecessary sensitive context. Across all evaluated methods, MINIM achieves the lowest TISL (0.101) and Keep% (12.0), demonstrating that task-conditioned contextual integrity training yields a strictly superior privacy–efficiency frontier relative to general instruction tuning.

**Discussion on Data Sparsity.** Although the average Node Exposure Ratio (Keep Rate) is approximately 12%, this reflects the sparsity of task-relevant elements in real-world accessibility trees. Typical DOM trees (often exceeding 500 nodes) are dominated by generic containers (e.g., <div>, group) and layout spacers with limited semantic contribution. A Keep Rate of $\approx 12\%$ therefore corresponds to filtering structural noise while retaining the functional elements (buttons, inputs, and task-relevant text) that support execution.

**Safety Validation.** Beyond aggregate metrics, we inspect the handling of specific high-risk injected elements. In diagnostic checks across the test set, MINIM suppresses $> 99.9\%$ of injected 2FA codes, passwords, and Slack notifications, consistent with the low TISL values reported above.

## 5.5. Task-Semantic Robustness

To verify that MINIM supports active agents beyond simple reading tasks, we stratify performance by task semantics in Table 3. The model maintains high utility across all categories, achieving perfect TCNP and TCNP-I (1.0000) on both *Transactional* (e.g., "Add to Cart", "Vote") and *Sensitive/Admin* (e.g., 2FA, Password) tasks, confirming that the minimization policy fully preserves essential affordances for complex interactive and security-critical flows. Table 3 further reveals the model's adaptive compression strategy. For **Sensitive/Admin** tasks, the model learns to retain a

larger fraction of nodes (Keep Rate: 15.84% vs. 8.66% for Informational), preserving sufficient context to support complex security flows without breaking task execution. This behavior emerges from the CI-aware objective, which encourages tighter minimization for low-stakes browsing while maintaining adequate structural coverage for high-stakes administrative actions.

*Table 3.* **Adaptive Minimization Strategy.** The model alters its compression rate and abstraction use based on task semantics, maintaining high utility across all categories. (TCNP: Context Recall; TCNP-I: Actionable Recall.)

| Task Semantics | K | A | TCNP | TCNP-I |
|---|---|---|---|---|
| Informational | 8.66% | 0.83% | 0.9156 | 0.9886 |
| Sensitive/Admin | 15.84% | 9.84% | 1.0000 | 1.0000 |
| Transactional | 10.88% | 3.52% | 1.0000 | 1.0000 |

## 5.6. Contextual Integrity Evaluation

We evaluate whether MINIM instantiates CI as an explicit information-flow rule rather than a purely conceptual framing. In MINIM, the task-conditioned necessity score specifies the *context*, the sensitivity score specifies the *information type*, and the transmission principle is implemented by the K/A/R policy. The central CI requirement in this setting is that disclosure is blocked whenever the task context renders it unnecessary, even if the information is otherwise benign.

**CI Compliance and Normative Bounds.** We benchmark against an Oracle policy that strictly removes all irrelevant nodes ($n_{i,T} \leq 1$), achieving 0.0% TISL by definition. On the full test set ($N = 662$), MINIM achieves TISL 0.101 (89.9% suppression relative to Full Observation) while maintaining a $\sim 12\%$ retention rate (K + A). This closely approximates the Oracle's strict "Need-to-Know" gate.

**Counterfactual Context Stress Test.** To test context dependence directly, we perform a counterfactual intervention: for a fixed subset of test instances, we keep the accessibility tree and node content unchanged but replace the goal input with a low-commitment *browse mode* context. Under this intervention, predicted necessity for task targets drops substantially for previously critical action nodes (e.g., checkout or submit controls), while irrelevant high-sensitivity content remains stably suppressed (predominantly R), indicating norm robustness under context shifts. For high-sensitivity elements that become only marginally useful under *browse mode*, the policy tends to favor A over R, preserving structural affordances without exposing raw values, consistent with proportional transmission. However, because R is triggered by a fixed necessity threshold ($\tau_{\text{nec}} = 1.0$), removals are less elastic than the underlying necessity scores under counterfactual contexts. A natural extension is dy-

| Policy Variant | Strategy | TCNP ↑ | TCNP-I ↑ | TISL ↓ |
|---|---|---|---|---|
| MINIM (Ours) | Adaptive | 0.9491 | 0.9931 | 0.1010 |
| Conservative | Map A →R | 0.9050 | 0.9650 | 0.0989 |
| Radical | Map A →K | 0.9491 | 0.9931 | 0.1014 |

*Table 4.* **Value of Abstraction.** Variants where the A action is forced to R (strict redaction) or K (full exposure). Residual leakage ($p_{abs} = 0.05$) is included in TISL for A actions. (TCNP: Context Recall; TCNP-I: Actionable Recall.)

namic thresholding or context-adaptive calibration to further tighten transmission norms in low-utility settings.

## 5.7. Ablation Studies

**Value of Adaptive Abstraction.** A key design choice in MINIM is the A action, which preserves structural affordances (e.g., buttons and inputs) while masking sensitive values. Table 4 reports two counterfactual variants: *Conservative (Privacy-First)* maps all A decisions to R, simulating strict redaction that blocks sensitive elements entirely and reducing TCNP-I by 2.81 absolute points (0.9931→0.9650) from losing critically mediated inputs (e.g., 2FA fields and address forms); *Radical (Utility-First)* maps all A decisions to K, recovering full actionability at a marginal leakage cost (TISL 0.1014 vs. 0.1010). Crucially, comparing MINIM with the Conservative baseline reveals the value of adaptive abstraction. Even when accounting for a 5% residual re-identification risk ($p_{abs} = 0.05$) in the privacy metric, MINIM incurs only a marginal increase in leakage (0.0989 → 0.1010) compared to the strict removal policy. In exchange for this small privacy cost, it recovers 2.81 absolute points in TCNP-I (0.9650→0.9931) and 4.41 in TCNP (0.9050→0.9491). This confirms that binary controls (Allow/Deny) are insufficient for agentic privacy: strictly denying sensitive nodes breaks task execution, while unconditionally allowing them elevates privacy risk; semantic abstraction provides the necessary intermediate option.

**Threshold Robustness.** As shown in Figure 2, sweeping $\tau_{nec}$ traces a stable privacy–utility frontier. Performance remains consistent across $\tau_{nec} \in [0.8, 1.5]$, suggesting that deployment does not require precise per-domain calibration.

## 6. Discussion

**Decoupling scoring from enforcement.** MINIM cleanly separates *contextual scoring* (predicting sensitivity and task-conditioned necessity) from *policy enforcement* (mapping scores to K/A/R decisions and rendering). This separation improves interpretability and portability: privacy–utility trade-offs can be adjusted by changing thresholds or the policy layer without retraining the scorer. Moreover, the realization of A is orthogonal to the scoring model. While we use standardized masking (e.g., [REDACTED]) for controlled evaluation, A can be implemented with richer sanitization

mechanisms such as typed placeholders, format-preserving redaction, or learned rewriting modules.

**Generalization and task coverage.** Our WebArena instantiation uses 27 curated task templates to support consistent node-level necessity annotation. This should be viewed as a controlled specification for evaluation rather than open-ended task coverage. Importantly, MINIM is not a task classifier; the task description serves as a conditioning signal for element-level necessity prediction, and we evaluate generalization across unseen pages, injected contexts, and domains. Extending to additional intents is primarily a data and specification problem: new tasks can be incorporated by collecting additional $(X, T)$ pairs under the same scoring rubric, without changing the framework. More broadly, although we evaluate on accessibility trees, the pre-disclosure minimization principle is representation-agnostic and applies to other structured observation channels (e.g., DOM trees, scene graphs, and tool schemas) where action-critical structure must be preserved while task-irrelevant sensitive content is withheld.

**Limitations.** Our threat model targets step-wise minimization against honest-but-curious remote inference. We do not address active adversaries (e.g., prompt injection) or cumulative privacy loss across long-horizon episodes. In addition, scaling to highly specialized enterprise interfaces or full desktop environments may require more efficient tree encoders and caching to handle larger structured observations under tight latency budgets.

## 7. Conclusion

We introduced MINIM, a framework that addresses Semantic Over-Privileged Observation in agentic systems. Grounded in Contextual Integrity, MINIM learns to distinguish task-critical affordances from task-irrelevant sensitive content and enforces a need-to-know disclosure rule over accessibility trees. Our dual-score formulation substantially reduces unnecessary sensitive exposure while preserving actionable utility, showing that strong privacy guarantees can coexist with reliable agent actuation. Although demonstrated on accessibility trees, the CI-driven structural scoring mechanism is representation-agnostic and extends to other structured observation formats such as DOM/VDOM, scene graphs, and tool-use schemas. Future work will study temporal privacy accounting over long-horizon interaction and multi-agent settings.

## Software and Data

We release the MINIM codebase in our GitHub repository. The repository includes the implementation, preprocessing and evaluation scripts, and instructions for reproducing our

experiments using the processed WebArena-derived structured observations and released sample data.

## Acknowledgements

This work was supported in part by the Office of Naval Research under grants N00014-24-1-2730 and N000142412663, Army Research Office under grant W911NF-24-1-0155, and the National Science Foundation under grants 2433904, 2247560, 2154929, 2235232, 2238635, 2154930, 2403758 and 2312447.

## Impact Statement

This work aims to improve privacy for autonomous agentic systems by minimizing structured observations before they are sent to remote inference. If adopted, it could reduce inadvertent exposure of sensitive information during routine agent interactions and enable safer deployment in privacy-sensitive settings. Potential risks include misuse to conceal information from oversight and uneven protection due to dataset or rubric bias; we view MINIM as a privacy-mitigation component rather than a complete security solution, and encourage future work on adversarial robustness and broader evaluation.

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

# A. Theoretical Formulation

We formalize the semantic minimization problem through the lens of Information Theory, providing the motivation for our CI-aware loss function (Section 4.2).

## A.1. Problem Statement

Let $X_t$ denote the full accessibility tree at time $t$, $T$ the user task, and $A^*$ the optimal agent action. The goal of MINIM is to learn a sanitization function $f(X_t, T) \rightarrow Z_t$ that produces a minimal observation $Z_t$.

We frame this as a **Constrained Contextual Bottleneck** problem. We seek to maximize the mutual information between the sanitized view $Z_t$ and the optimal action $A^*$, subject to a constraint on the leakage of sensitive attributes $S$:

$$\max_f \ I(Z_t; A^*|T) \quad \text{s.t.} \quad I(Z_t; S|T_{\text{irr}}) \leq \epsilon \tag{5}$$

where $T_{\text{irr}}$ denotes contexts where $S$ is task-irrelevant ($n_{i,T} = 0$).

## A.2. Loss Function Derivation

Our dual-score framework serves as a tractable proxy for this objective:

- **Necessity** $\hat{n}$ approximates *action relevance* $P(\text{Actionable}|X_t, T)$, acting as a gate for $I(Z_t; A^*)$.

- **Sensitivity** $\hat{s}$ approximates the inherent risk $P(\text{Sensitive}|X_t)$, quantifying the cost in the leakage constraint.

The CI-weighted necessity loss (Eq. 6) implements the Lagrangian relaxation of this constrained optimization problem.

$$\mathcal{L}_{\text{nec}}^{\text{CI}} = \sum_{i,t} \underbrace{\left(1 + \lambda \cdot \frac{s_i}{10} \cdot \left(1 - \frac{n_{i,T}}{10}\right)\right)}_{\text{Lagrange Multiplier Proxy}} \cdot |\hat{n}_{i,T} - n_{i,T}| \tag{6}$$

The term $\lambda \cdot s_i \cdot (1 - n_{i,T})$ acts as a dynamic penalty coefficient that becomes large strictly when privacy risk is high ($s \rightarrow 10$) and utility is low ($n \rightarrow 0$), aligning the gradient descent direction with the minimization of $I(Z_t; S|T_{\text{irr}})$.

# B. Implementation Details

## B.1. Policy Logic

The Normative Policy Layer maps predicted scores $(\hat{s}_i, \hat{n}_{i,T})$ to disclosure actions via thresholds $\tau_{\text{nec}} = 1.0$ and $\tau_{\text{sens}} = 5.0$. This threshold-based logic separates the *estimation* of risk (model output) from the *decision* of acceptable risk (policy), allowing administrators to adjust $\tau$ post-deployment without retraining.

## B.2. Model Details

We use a local deployable Graph Attention Networks v2 (GATv2) over the accessibility tree. The raw tree is converted into a graph where each UI element is represented as a node with a 512-d feature vector constructed from a 384-d MiniLM text embedding together with UI attributes and structural features, which, along with the tree edges, serves as the input to the GATv2 model. The backbone consists of 3 GATv2 layers (hidden size 256, 4 heads) capturing topological dependencies across UI elements. Outputs are passed to two MLP heads ($256 \rightarrow 128 \rightarrow 11$) for sensitivity and task-conditioned necessity prediction.

## B.3. Training Setup

Table 5 outlines the specific hyperparameters used for training. We use $\alpha = 1.0$ and $\lambda = 1.0$ to balance the multi-task objective.

---

**Algorithm 1** MINIM Client-Side Sanitization

---

**Input:** Accessibility Tree $X_t = \{e_i\}_{i=1}^N$, Task $T$
**Parameters:** Model $\theta$, Thresholds $\tau_{\text{nec}}, \tau_{\text{sens}}$
**Output:** Sanitized Tree $Z_t$
$Z_t \leftarrow []$
$H \leftarrow \text{Encoder}_\theta(X_t, T)$
**for** $i = 1$ **to** $N$ **do**
   $\hat{s}_i, \hat{n}_{i,T} \leftarrow \text{Heads}_\theta(H_i)$
   **if** $\hat{n}_{i,T} < \tau_{\text{nec}}$ **then**
     **continue** {Apply R }
   **else if** $\hat{s}_i \geq \tau_{\text{sens}}$ **then**
     $e_i' \leftarrow \text{Abstract}(e_i)$
     $Z_t.\text{append}(e_i')$ {Apply A }
   **else**
     $Z_t.\text{append}(e_i)$ {Apply K }
   **end if**
**end for**
**return** $Z_t$

---

*Table 5.* **Training Hyperparameters.** Fixed values for reproducibility.

| Hyperparameter | Value |
|---|---|
| Learning Rate | $3 \times 10^{-4}$ |
| L2 Regularization | $1 \times 10^{-5}$ |
| Optimizer | AdamW |
| Batch Size | 32 |
| Epochs | 10 |
| CI-Weight ($\lambda$) | 1.0 |
| Loss Balance ($\alpha$) | 1.0 |

## C. Dataset Components and Statistics

### C.1. Data Composition

We construct a privacy-augmented corpus from WebArena (Zhou et al., 2024), spanning Shopping, Reddit, and Gmail. To simulate realistic risks, we inject synthetic sensitive content (e.g., 2FA codes, emails) into standard accessibility trees (Table 6).

*Table 6.* Injected sensitive content categories.

| Category | Example | $s_i$ **Range** |
|---|---|---|
| 2FA/OTP codes | "Verification code: 847293" | 10 |
| Password prompts | "Save password..." | 10 |
| Slack notifications | "Alice: Can you review..." | 8–9 |
| Email previews | "From: json@..." | 7–9 |

### C.2. Statistical Characteristics

Table 7 summarizes the dataset scale and the distribution of privacy conflicts (nodes that are both Sensitive and Irrelevant).

## D. Additional Empirical Results

We provide a domain-level breakdown of model performance to scrutinize reliability across different web environments.

*Table 7.* **Dataset Overview.** Left: Global statistics. Right: Conflict analysis of training nodes ($N \approx 5.4M$).

| Metric | Value |
|---|---|
| Unique Trees | 150 |
| Task Types | 27 |
| **Total Variants** | **5,403** |
| Training Variants | 4,741 |
| Test Variants | 662 |

| Condition | Count | % |
|---|---|---|
| Irrelevant (Nec$\leq$1) | 4.6M | 85.9% |
| High Sens (Sens$\geq$5) | 164k | 3.0% |
| **Risk** (Irr $\wedge$ Sens) | 115k | **2.1%** |

*Table 8.* **Top Interface Roles and Average Scores.** Common UI elements exhibit distinct necessity/sensitivity profiles.

| Role | Freq | Avg Nec | Avg Sens |
|---|---|---|---|
| InlineTextBox | 26.8% | 0.01 | 0.08 |
| none | 21.0% | 0.44 | 0.00 |
| StaticText | 17.7% | 0.05 | 0.11 |
| generic | 6.1% | 0.53 | 0.02 |
| link | 4.7% | 4.64 | 0.38 |
| button | 3.3% | 4.30 | 0.31 |
| listitem | 3.1% | 1.15 | 0.17 |
| list | 2.0% | 1.15 | 0.00 |

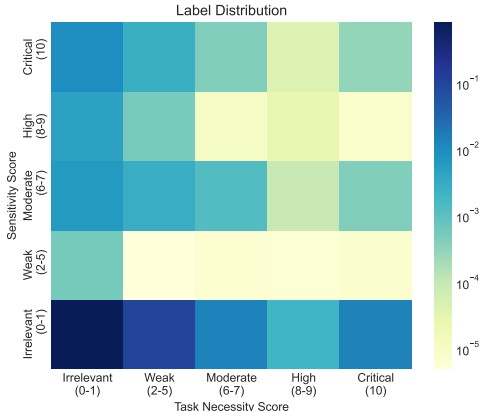

*Figure 4.* Joint distribution of Necessity vs. Sensitivity scores. Empty bins reflect discrete annotation rubrics.

*Table 9.* **Comprehensive Task Suite.** The evaluation dataset covers 27 distinct task categories across three domains, spanning informational, transactional, and diagnostic scenarios.

| Domain | Tasks and Objectives |
|---|---|
| **Shopping** | *search_product, view_product_details, add_to_cart, view_cart, read_reviews, compare_products, add_to_wishlist, access_bookmark, switch_tab, extract_2fa_code, complete_2fa* |
| **Reddit** | *read_post, read_comments, vote_content, search_discussion, navigate_forums, reply_comment, open_slack_app* |
| **Gmail** | *read_email, compose_email, reply_email, search_email, forward_email, delete_email, save_password, read_slack_message, reply_slack* |

*Table 10.* **Domain-Specific Performance.** Breakdown of context recall (TCNP) and privacy (NORMTISL) on the held-out test set. Reddit exhibits higher residual leakage due to the semantic ambiguity of user-generated content. (TCNP: Context Recall.)

| Domain | Test $N$ | Context Recall (TCNP) $\uparrow$ | Privacy (TISL) $\downarrow$ |
|---|---|---|---|
| Gmail | 472 | 96.61% | 1.88% |
| Reddit | 130 | 95.38% | 13.50% |
| Shopping | 60 | 93.60% | 0.85% |

As shown in Table 10, we observe distinct TISL profiles across domains, while context recall (TCNP) remains consistently high (>93%) in all three. Reddit exhibits the highest residual leakage (13.50%), consistent with the semantic ambiguity of user-generated content where the boundary between "public discussion" and "private identifier" is less structurally distinct. Shopping and Gmail achieve substantially lower leakage (0.85% and 1.88%, respectively), reflecting the model's ability to stably remove task-irrelevant sensitive content on structured task pages and email interfaces.

## E. Baseline Evaluation Prompts

To ensure reproducibility, we provide the exact system prompts used to query the open-weight LLM baselines (Qwen3-8B-Instruct, Nemotron-Nano-9B, GPT-OSS-20B, Llama-3.3-70B-Instruct, Mistral-7B-v0.3, Llama-3-8B-Instruct, and Gemma-3N-E4B) for the zero-shot scoring evaluation. These prompts implement the Contextual Integrity-aligned scoring rubric described in Section 4.

### SYSTEM PROMPT: TASK-CONDITIONED NECESSITY SCORING

```
You are a strict scoring judge for accessibility-tree nodes, focusing ONLY on TASK-CONDITIONED NECESSITY.

Your task:
For each provided node, assign ONE integer necessity score:
- necessity_score n in {0,1,2,3,4,5,6,7,8,9,10}

Scores MUST be integers. Do NOT use ranges or decimals.
Use ONLY the provided node fields and local context. Do NOT assume hidden UI.

Return JSON only. No markdown. No code fences. No extra keys.

Output schema (strict):
{
  "results": [
    {
      "nodeId": "string",
      "necessity_score": 0-10 integer,
      "necessity_reason": ["short_reason", "..."],
      "confidence": 0.0-1.0 number or null
    }
  ]
}

-----------------------------------------------------------------
TASK-CONDITIONED NECESSITY SCORING RUBRIC (n) -- INTEGER ANCHORS
-----------------------------------------------------------------
n = 0
- Completely irrelevant to the task; removing has zero impact.
- Examples: ads, footer/legal boilerplate, unrelated feed items, decorative regions.

n = 1
- Pure noise or redundant UI; visible but provides no help even indirectly.
- Examples: repeated icons, empty text nodes, spacer elements.

n = 2
- Extremely weak contextual presence; could be noticed but does not aid task completion.
- Examples: generic counts/metrics for unrelated content, non-task badges.

n = 3
- Marginally related context; slight orientation benefit but task unaffected without it.
- Examples: post metadata, author labels when the task is navigation.

n = 4
- Supportive but optional; helps orientation or confidence but not required.
- Examples: section headings, explanatory labels, non-essential descriptions.

n = 5
- Moderately helpful; improves efficiency/understanding but alternatives exist.
- Examples: secondary navigation links, optional filters/sort controls.

n = 6
- Strongly helpful; reduces friction or steps, but not strictly required.
- Examples: sidebar community list when search/navigation alternatives exist.

n = 7
- Important task enabler; most users rely on this to complete the task.
- Examples: primary navigation menu for the task, main forum/subreddit list entry points.
```

```
n = 8
- Core task component; task becomes very difficult without it.
- Examples: the specific forum/subreddit link needed to enter a forum.

n = 9
- Near-critical; task almost certainly fails without it.
- Examples: the only visible control to switch/select forums in the provided context.

n = 10
- Absolutely required; task is impossible without it given the provided context.
- Examples: sole entry point to forum navigation or the only actionable control required.

------------------------------------------------------------------
GENERAL RULES
------------------------------------------------------------------
- Judge necessity strictly in the context of the given task.
- If uncertain between two adjacent scores, choose the LOWER one and explain briefly why.
- Reasons should be short tags/phrases (<= 8 items), grounded in node/context fields.
- Baseline scores may be incorrect; judge independently.
```

## SYSTEM PROMPT: SENSITIVITY SCORING

```
You are a strict scoring judge for accessibility-tree nodes, focusing ONLY on SENSITIVITY (privacy / secrets /
     PII).

Your task:
For each provided node, assign ONE integer sensitivity score:
- sensitivity_score s in {0,1,2,3,4,5,6,7,8,9,10}

Output schema (strict):
{
  "results": [
    {
      "nodeId": "string",
      "sensitivity_score": 0-10 integer,
      "sensitivity_reason": ["short_reason", "..."],
      "confidence": 0.0-1.0 number or null
    }
  ]
}

------------------------------------------------------------------
SENSITIVITY SCORING RUBRIC (s) -- INTEGER ANCHORS
------------------------------------------------------------------
s = 0
- Purely structural or decorative; no content and no identifying information.
- Examples: layout containers, separators, unlabeled icons.

s = 1
- Generic public UI labels/controls; not user-specific or session-specific.
- Examples: "Home", "Search", "Submit", "Upvote", "Next".

s = 2
- Public content with semantic meaning but clearly non-personal; safe verbatim.
- Examples: generic navigation headings, subreddit names, non-personal post titles.

s = 3
- Mild contextual info that could be weakly identifying only in aggregate; not tied to an individual.
- Examples: generic timestamps, popularity metrics, generic categories.

s = 4
- Public user-related identifiers; enables lightweight profiling but not private identification.
- Examples: public usernames/handles, public participation labels.

s = 5
- Behavioral or preference signals tied to a user account; supports profiling across contexts.
- Examples: joined/subscribed communities, voting history indicators, personalized recommendations.

s = 6
- Semi-sensitive personal attributes (not secrets) that may cause discomfort or privacy loss.
- Examples: inferred interests, activity summaries, account status indicators.

s = 7
- Direct personal data (non-secret) that clearly identifies a person.
- Examples: real name, email address, phone number.

s = 8
```

```
 - Highly sensitive personal data; serious privacy risk if exposed.
 - Examples: precise home address, private messages, financial identifiers.

s = 9
 - Security-critical or authentication-related data; exposure may enable compromise.
 - Examples: password fields, 2FA/OTP codes, recovery keys.

s = 10
 - Extremely sensitive secrets or irreversible compromise material.
 - Examples: plaintext passwords, private cryptographic keys, API tokens/secrets.
 -------------------------------------------------------------------
```

