# OpenReview forum: "MINIM: Privacy-Aware Minimal View for Agents via Trusted Local Sanitization"
_ICML.cc/2026/Conference — ICML 2026 regular_

### Official Review · Reviewer_w9tr · 2026-03-09

**Soundness:** 2
**Presentation:** 3
**Significance:** 2
**Originality:** 2
**Overall Recommendation:** 3
**Confidence:** 3

**Summary:**

This paper introduces MINIM, a trusted local broker designed to mitigate "Semantic Over-Privileged Observation" in LLM-powered agents by performing privacy-aware minimization on structured UI observations before they are transmitted to remote servers. Grounded in Contextual Integrity (CI), the framework utilizes a specialized local model to predict both an inherent sensitivity score ($s$) and a task-conditioned necessity score ($n$) for each UI element. These dual scores drive a trinary disclosure policy—Keep, Abstract, or Remove—that is optimized through a CI-aware objective penalizing the disclosure of high-risk, low-utility content. Experimental results on a privacy-augmented WebArena dataset show that MINIM reduces task-irrelevant sensitive leakage to 1.1% while maintaining 95.7% of actionable utility, effectively outperforming zero-shot prompted LLM baselines such as Llama-3 and Qwen-3.

**Compliance With Llm Reviewing Policy:**

Affirmed.

**Key Questions For Authors:**

1. The proposed loss functions lack theoretical guarantee and convergence analysis.
2. The dataset construction is relatively simple and lacks diversity in real-world scenarios.
3. In addition, the paper lacks necessary implementation details.

More detail please refer to weakness.

**Limitations:**

refer to weakness.

**Strengths And Weaknesses:**

**Strength:**

1. The paper addresses a critical and increasingly relevant security gap in autonomous agent systems—Semantic Over-Privileged Observation. As LLM-powered agents rely more heavily on rich UI observations to function, the risk of leaking sensitive context (such as authentication codes or private notifications) to remote servers becomes a primary concern for deployment. This work provides a necessary framework for privacy-aware agentic control.

2. Unlike prior defenses that focus on unstructured text or pixel-based inputs, this work innovates by utilizing structured accessibility trees as the primary basis for privacy analysis. By treating the UI as a hierarchy of discrete elements, the authors can implement a learned structural bottleneck. This allows for fine-grained, task-conditioned minimization that preserves the agent's interactive affordances while suppressing irrelevant sensitive attributes.

3. The manuscript is logically organized and the technical presentation is clear and easy to follow.


**Weaknesses:**
1. While the paper mentions several related agent-privacy frameworks such as AirGapAgent, Portcullis, and Papillon , the experimental evaluation in Table 1 and Table 2 primarily compares MINIM against "Fixed Policies" (e.g., Sensitivity-Only) or zero-shot prompted LLMs. The lack of a direct empirical comparison with state-of-the-art specialized privacy gateways makes it difficult to assess MINIM's relative advancement in the field.

2. The proposed CI-aware necessity loss introduces a complex weighting mechanism based on ground-truth sensitivity ($s_i$) and necessity ($n_{i,T}$). The authors do not provide a formal convergence proof to demonstrate that this objective is theoretically sound or stable across different initializations. Furthermore, it is unclear how the model effectively converges when faced with varying hyperparameter settings, particularly given the lack of a detailed parameters sensitivity analysis .

3. The manuscript refers to a "specialized local model" and a "Tree Encoder" but fails to provide specific design details. Essential information regarding the model's architecture, such as whether it is based on GNNs, Transformers, or MLPs. As well as its layer count and embedding dimensions, is missing.

4. The evaluation relies on the WebArena dataset with "synthesized system-level sensitive context" (e.g., 2FA codes and password prompts) injected during preprocessing. This synthetic injection approach may not accurately capture the nuanced and messy privacy leakage patterns found in real-world, non-templated digital environments. Consequently, the high performance reported might be failed to reflect a real-world scenorias.

---

> ### Author Rebuttal · Authors · 2026-03-31
>
> We appreciate the reviewer’s comments and for recognizing the importance of Semantic Over-Privileged Observation and the novelty of leveraging structured UI trees.
>
>
> **1. Comparison with SOTA Privacy Frameworks (W1):**
> We appreciate the reviewer raising this point.
>
> A direct comparison with prior frameworks (e.g., AirGapAgent, Portcullis) is challenging due to fundamental differences in **input representation** and **policy objectives**.
>
> - **Representation mismatch:** Existing methods are designed for **flat natural language inputs (prompt-level)**, whereas MINIM operates on **structured accessibility trees** with explicit hierarchical and relational information. These frameworks cannot be directly applied to node-level structured observations with graph dependencies.
> - **Different privacy objectives:** Prior approaches typically focus on **universal PII removal**, while MINIM follows the **Contextual Integrity** principle, preserving sensitive information only when it is necessary for the task.
>
> As a result, these methods cannot be evaluated under the same input setting or objective, making a fair quantitative comparison infeasible.
>
> We view MINIM as **complementary** to such approaches: prior work sanitizes textual prompts, while MINIM minimizes structured system-level observations before they are exposed to the agent.
>
>
>
> **2. Theoretical Guarantee and Convergence (W2/Q1):**
> We thank the reviewer for this important question.
>
> Our method is grounded in a formal **Contextual Integrity (CI)** formulation, where privacy is defined as restricting inappropriate information flow under task context. Based on this principle, we formulate structured observation minimization as a **privacy–utility trade-off optimization problem**, and derive the CI-aware necessity loss as a principled objective. In this formulation, sensitivity serves as a proxy for disclosure risk and modulates the penalty on necessity errors precisely in high-risk regimes.
>
> We would like to clarify that our primary contribution lies in this **CI-based problem formulation and objective design**, rather than proposing a new optimization algorithm. The resulting objective defines a structured trade-off between utility and privacy, which is optimized using standard training procedures.
>
> Empirically, we find that performance is not highly sensitive to moderate hyperparameter variations, and thus report a representative and stable configuration in Appendix B. We will include additional ablation and sensitivity analysis in the extended appendix of the final version.
>
>
> **3. Implementation Details and Model Architecture (W3/Q3):**
>
> For clarity, MINIM employs a lightweight relation-aware **GATv2** model over accessibility trees, where nodes correspond to UI elements and edges encode structural relations (e.g., parent–child and sibling). Each node is initialized with a compact **MiniLM** embedding combined with UI attributes and structural features. The graph backbone consists of three GATv2 layers (hidden size 256, 4 attention heads), producing contextualized node representations for downstream scoring. The graph module contains ~1.5M trainable parameters, enabling efficient structured reasoning with low overhead.
>
> The training configuration (e.g., optimizer, learning rate, and regularization) is provided in Table 5, Appendix B. We use standard hyperparameter settings selected through routine validation, yielding stable performance and low sensitivity to moderate variations. We will further expand architectural and training details (including hyperparameter sensitivity analysis) to improve clarity and reproducibility.
>
>
>
>
> **4. Synthetic Dataset Realism(W4/Q2):**
>  Our goal is not to construct a fully comprehensive benchmark of real-world leakage scenarios, but to enable **controlled and measurable evaluation** of task-conditioned privacy decisions.
>
> Our setup is built on **real accessibility trees from WebArena**, and the injected sensitive content is **structure-preserving**: it follows the same schema as native UI elements (e.g., text fields, notifications, or input nodes) without introducing artificial patterns. As a result, injected elements are structurally indistinguishable from existing UI content, requiring the model to reason over realistic UI structure rather than synthetic artifacts.
>
> Synthetic injection is necessary because real-world datasets with **reliable, task-conditioned privacy annotations** are not available, and collecting such data raises ethical concerns. This approach allows us to systematically evaluate the privacy–utility trade-off under well-defined conditions.
>
> More broadly, MINIM is a framework that can be adapted to new domains by constructing data under the corresponding UI schema and privacy policies, rather than being tied to a specific dataset.

---

> > ### Author Rebuttal · Reviewer_w9tr · 2026-04-03
> >
> > While leveraging the Accessibility Tree is a novel concept, the methodology and experimental results do not provide strong enough evidence to support the design's effectiveness.
> >
> > 1. The authors’ argument that comparisons are "infeasible" due to different input formats is unconvincing. Since the core privacy objective remains the same—including "universal PII removal"—the authors could have adapted existing baselines to demonstrate MINIM’s relative advantage.
> >
> > 2. The heavy reliance on synthetic "toy examples" (WebArena) is a significant weakness. Without testing on diverse, real-world UI environments, the practical value of this tool remains unproven. Even though the authors claim that "MINIM is a framework that can be adapted to new domains by constructing data under the corresponding UI schema and privacy policies," the current dataset is still insufficient to prove its effectiveness.
> >
> > Thanks for your arguments. I keep my original score.

---

> > > ### Author Response · Authors · 2026-04-03
> > >
> > > We thank the reviewer for the continued feedback and would like to clarify this point more precisely.
> > >
> > > While prior works (e.g., AirGapAgent, Portcullis, PAPILLON) aim to reduce sensitive information exposure, they address a **different problem setting**. These approaches are designed for **prompt- or conversation-level inputs**, where the objective is to remove or rewrite sensitive content whenever possible. In contrast, MINIM is designed to **retain sensitive information when it is necessary for successful task execution**. Moreover, prior methods typically produce textual outputs evaluated at the language level, whereas our setting requires preserving **structured representations that remain directly actionable by an agent**.
> > >
> > >
> > > This difference arises from the **underlying decision problem**, rather than merely the input format. Prior methods make decisions over **text spans or tokens**, whereas MINIM operates on **structured UI observations**, where each element corresponds to an **actionable component** within the interface. As a result, decisions must account not only for **sensitivity**, but also for whether the element is **required for the agent to act**.
> > >
> > > In this setting, **removing sensitive information is not always valid**, as certain elements are **necessary for completing the task**.
> > > For example:
> > > - **Login workflows** require one-time passwords (OTP), which are sensitive but essential.
> > > - **Form interactions** require fields such as addresses, accurate credit card PINs, or account identifiers.
> > > - **Decision interfaces** (e.g., submit payment) depend on attributes such as credit card numbers or billing addresses.
> > >
> > > Applying prompt-level sanitization in this context would either (i) remove such elements and break the action sequence, or (ii) retain them without structured control, leading to unnecessary leakage. MINIM instead explicitly models the **trade-off between necessity and sensitivity** under task constraints.
> > >
> > > Importantly, these prior methods address only part of this problem. They can be applied after the relevant content has been determined (e.g., for rewriting or abstraction), but do not address the **earlier decision** of whether an element should be kept, abstracted, or removed under task context.
> > >
> > > Regarding evaluation, our setup focuses on the **step-wise decision setting** in which the agent acts on the current UI state at each step. We adopt this level of evaluation because it directly measures whether the **structured observation preserves the information and affordances required for action**.
> > >
> > > We therefore view these approaches as **complementary** rather than directly comparable, and believe our formulation reflects the requirements of **structured, task-conditioned agent settings**.

---

### Official Review · Reviewer_5ENT · 2026-03-12

**Soundness:** 2
**Presentation:** 2
**Significance:** 2
**Originality:** 2
**Overall Recommendation:** 4
**Confidence:** 3

**Summary:**

This paper addresses the "Semantic Over-Privileged Observation" problem in LLM-powered agents, where full UI states are transmitted to remote servers, leading to privacy leaks. The authors propose MINIM, a trusted local broker that uses a dual-score model to predict the sensitivity and task-conditioned necessity of UI elements. Grounded in Contextual Integrity, it implements a trinary policy, “Keep, Abstract, or Remove”, to sanitize observations before disclosure. Experiments on a privacy-augmented WebArena dataset show that MINIM significantly reduces sensitive leakage while maintaining high actionable utility.

**Compliance With Llm Reviewing Policy:**

Affirmed.

**Key Questions For Authors:**

(1) How does the performance of MINIM scale with the size of the accessibility tree, especially in complex enterprise applications with thousands of nodes?

(2) What is the specific architecture of the specialized local model used for scoring? Is it lightweight enough for mobile or edge devices?

(3) Could you clarify how Abstraction is implemented for non-textual elements, such as images that contain sensitive visual information?

**Limitations:**

Yes

**Strengths And Weaknesses:**

Strengths

(1) The conceptual framework of decoupling sensitivity from necessity is intuitive and addresses the dynamic nature of privacy in agentic tasks.

(2) Grounding the system in Contextual Integrity provides a theoretical foundation for why certain information should or should not flow.

Weaknesses

(1) While WebArena is a standard benchmark, the 27 task templates might limit the diversity of necessity judgments. It is unclear how the system would handle highly ambiguous or cross-application user intents.

(2) Reliance on a specialized local model: The performance of MINIM heavily depends on the local scoring model's accuracy. The paper could further discuss the computational overhead and latency of running such a model locally in real-time control loops.

(3) While the authors highlight that the scoring mechanism is representation-agnostic , the current experiments are centered on accessibility trees. Demonstrating performance on a different structured modality (e.g., scene graphs) would further validate the framework's broad applicability.

---

> ### Author Rebuttal · Authors · 2026-03-31
>
> We thank the reviewer for the thoughtful feedback and for recognizing the conceptual framework and theoretical grounding of our approach.
>
> **1. Latency and Scalability (W2/Q1):**
> To empirically address concerns regarding real-time viability, we conducted additional benchmarking during this rebuttal period. On a standard laptop (Apple MacBook Pro, M1 CPU), the complete inference pipeline (MiniLM text encoding + GATv2 scoring) processes an accessibility tree (averaging $\sim$ 450 nodes) in **$\sim$ 500 milliseconds (0.5s)** without hardware acceleration.
> Because the underlying architecture follows an **$O(N)$ linear complexity**, this sub-second algorithmic overhead remains practically negligible even when scaling to **thousands of nodes** in more complex applications. Compared to the multi-second generation latencies inherent to remote LLM agents (e.g., GPT-4V or LLaMA-based solvers), MINIM adds negligible delay, ensuring it functions as a frictionless local broker strictly suited for real-time control loops.
>
> While our measurement isolates the efficiency of the perception component (0.5s/step), we recognize that end-to-end performance in multi-step agentic tasks involves broader system factors (e.g., planning and environment feedback). We believe profiling these full system-level interactions is a valuable direction for a dedicated study.
>
>
> **2. Task Diversity and Generalization (W1):**
> We thank the reviewer for raising this important point.
> While we use 27 templates for controlled evaluation, MINIM does not "memorize" task patterns but learns a **principled functional mapping** between (Task Intent, UI Context) and (Sensitivity, Necessity).
>
> - **Handling Ambiguity:** For underspecified or ambiguous intents, our dual-scoring mechanism acts as a **fail-safe**. Because sensitive elements carry an inherent risk score, the model defaults to a conservative policy (Abstract or Remove) unless a strong, task-specific necessity signal is established.
> - **Cross-Application Support:** Since accessibility trees follow **standardized system schemas** (e.g., standard roles and interactive flags) across different platforms, the model’s structural reasoning naturally generalizes to multi-application scenarios.
> - **Framework Adaptability:** Rather than assuming a single model generalizes to all scenarios, MINIM is designed to be extended to new tasks through policy-driven data generation. For a new application setting, a corresponding privacy policy can be specified to construct task-conditioned training data, after which the model can be fine-tuned to capture the appropriate disclosure behavior. This enables systematic adaptation to new domains without relying on fixed patterns.
>
> **3. Specific Architecture and Mobile/Edge Viability (W2/Q2):**
> We used a lightweight relation-aware Graph Attention Network (GATv2) over the accessibility tree, combined with a compact MiniLM-based text encoder. The graph backbone has 1.5M trainable parameters, and MiniLM itself is a small, widely-used sentence encoder that has been successfully deployed in on-device and edge settings. This makes the overall model efficient and suitable for local deployment. The GATv2 architecture naturally captures the structured dependencies in accessibility trees (e.g., parent–child and sibling relations), enabling effective node-level reasoning over UI structure. We will provide additional implementation details in the revision.
>
>
> **4. Abstraction for Non-Textual Elements (Q3):**
> We clarify that our abstraction mechanism operates over **structured accessibility nodes rather than raw pixel segments**.
> - **Mechanism:** In modern OS accessibility layers, non-textual elements (images, icons) are represented as nodes with associated semantic metadata (e.g., "content descriptions" or "alt-text"). MINIM performs reasoning over these attributes within the global UI context.
> - **Redaction:** Abstraction is applied to these semantic fields. If an image is deemed sensitive and task-irrelevant, its labels are replaced with generic placeholders while preserving the node's **functional role and interactive affordances**. This ensures the agent can recognize an actionable object exists without accessing its underlying sensitive visual content.
> - **Scope:** While metadata-level abstraction is highly efficient for structured UI observations, we recognize that **content-level visual redaction** (e.g., pixel-level in-painting) is a distinct and valuable research area that warrants independent future study.

---

> > ### Author Rebuttal · Reviewer_5ENT · 2026-04-02
> >
> > Thanks for your response. I keep my original score.

---

### Official Review · Reviewer_vcJM · 2026-03-13

**Soundness:** 3
**Presentation:** 3
**Significance:** 4
**Originality:** 4
**Overall Recommendation:** 5
**Confidence:** 4

**Summary:**

The paper describes a method to ensure contextual integrity via data minimization in UI elements, by learning a local broker that sanitizes/abstracts elements away from a UI state prior to it being sent to inference servers. They compare against fixed policies and prompted LLM-based abstraction/element removal modules, and show that their method successfully outperforms these baselines while also showcasing the importance of abstracting away elements as opposed to removing them entirely.

**Compliance With Llm Reviewing Policy:**

Affirmed.

**Final Justification:**

After a discussion with the authors regarding my concerns (please refer to my rebuttal acknowledgement and the following discussion), all of the stated weaknesses and concerns stand satisfactorily addressed. With this, I recommend acceptance of this paper.

**Key Questions For Authors:**

Can you please address the stated weaknesses and provide clarifications or additional results wherever necessary?

**Limitations:**

Yes

**Strengths And Weaknesses:**

# Strengths
- [S1] Novel and convincing application of contextual integrity, especially in a time where browser agents and computer use agents have gained prominence and share UI elements with inference servers with potentially a lot of sensitive information.
- [S2] The evaluation metrics are well-defined and cover utility preservation (preservation of necessary nodes) and leakage of sensitive information.
- [S3] The authors study and provide ample empirical evidence and rationale for including an abstraction option rather than purely removing elements. If this was not included, I would have questioned the authors about it, so I commend the authors for including this study.
- [S4] The threshold is calibrated thoughtfully by performing a pareto front based selection
- [S5] The provided results on data minimization/abstraction while preserving utility are convincing, especially when compared against the reasonably chosen baselines (which I like) that practitioners might otherwise ordinarily use (LLM-based) or rule-based baselines that prioritize one aspect over another (privacy vs. utility) or are completely random.
- [S6] The discussion of the data corpus used in the experiments and the derivation of instances is very well and clearly described in the experimental setting subsection.
- [S7] The discussion about TISL reduction in "Utility vs. Privacy Trade-offs." is interesting and relevant to this setting involving rich UI states, and puts the observations into perspective. Similar in-depth insights are provided in "Discussion on Data Sparsity" (lines 357 - 366). Such insights are appreciated.

# Weaknesses
- [W1] The definition of the accessibility tree, which should be a graph with nodes and edges as it is a tree, seems incomplete. In particular, the edges are not defined; only the nodes are.
- [W2] The definition of $L_{nec}$ seems contrived and not sufficiently explained. In Equation 3, in the definition of necessity loss, why can't the loss here be only w.r.t. necessity and the overall loss an affine sum of the two simply defined losses ($L_{sens}$ and $L_{nec}$)? Why is the sensitivity risk needed here if the first loss term already handles it? Can the authors describe why they define $L_{nec}$ as such and how they arrived at it?
- [W3] [Lines 247-248] The authors report results on a test set of 662 instances out of a total of 5403 instances. Isn't that a nearly 88:12 train-test split? That seems data inefficient w.r.t. training data. Can MINIM work sufficiently well when trained with a more reasonable train-test split?
- [W4] Also, I am not sure if I am missing something, but while the objectives used for training the MINIM classifier is very well-defined, the choice of architecture/LLM is not specified.
- [W5] In Figure 2, why is the threshold set to 1.0? Wouldn't a threshold like $\tau_{nec}$ = 6 be better and yield the best (or close) utility with even lower sensitive data exposure?

Upon these weaknesses being appropriately addressed, I shall be glad to raise my score, as then the paper would be more sound and impactful in my opinion.

---

> ### Author Rebuttal · Authors · 2026-03-31
>
> We thank the reviewer for the thoughtful feedback and for recognizing the novelty of our work. We address the concerns below.
>
> **1. Accessibility Tree Edges (W1):**
> The accessibility tree is a **standard system-level representation**, not introduced by our work, widely used in modern UI frameworks (e.g., accessibility APIs and DOM structures). In this representation, nodes correspond to UI elements, and edges encode structural relationships such as **parent–child hierarchy and sibling relations**.
> Our method directly operates on this existing structure rather than redefining it. These structural edges provide the necessary context for modeling how elements relate within the interface, which is essential for task-conditioned necessity and sensitivity reasoning.
> We will further clarify this representation and provide additional background in the revision to avoid potential confusion.
>
> **2. Necessity Loss (Eq 3) and Sensitivity Risk (W2):**
> The key rationale is that necessity and sensitivity interact in a non-additive manner in privacy risk. A simple affine combination (e.g., $\mathcal{L}_{sens} + \mathcal{L}_{nec}$) treats them as separable objectives, and therefore cannot distinguish necessity errors on low-risk elements from those on highly sensitive ones.  For example, under a simple additive formulation, a necessity error on a benign element (e.g., a structural `<div>`) would be penalized identically to the same magnitude of error on a highly sensitive element (e.g., a 2FA code).
>
> From a **Contextual Integrity** perspective, these cases have fundamentally different implications. Our design explicitly models this **asymmetry**: incorrectly retaining task-irrelevant sensitive information represents a significantly higher privacy risk than errors on low-sensitivity elements.
>
> Importantly, sensitivity loss $\mathcal{L}_{\text{sens}}$  and the sensitivity term in Eq. 3 play distinct roles. The sensitivity loss supervises the accuracy of sensitivity prediction, but does not influence how necessity errors are penalized. The multiplicative factor introduces a **risk-aware coupling**, where necessity errors are dynamically weighted by sensitivity, increasing penalties precisely in high-risk regimes.
>
> Therefore, the sensitivity term is not redundant with $\mathcal{L}_{sens}$, but serves as a **dynamic reweighting mechanism** that aligns optimization with minimizing high-risk disclosures, rather than uniformly minimizing prediction error.
>
>
>
>
> **3. Train/Test Split (W3)**
> We thank the reviewer for raising this point. The 4,741/662 split is not manually chosen or tuned, but follows directly from our data collection pipeline, where a disjoint subset of task instances is held out for evaluation.
>
> We agree that data efficiency is an important consideration. While performance naturally benefits from more training data, each accessibility tree contains hundreds of UI elements, providing rich structured signals for learning task-conditioned behavior. As a result, the effective supervision is not limited to the number of trees alone.
>
> We expect MINIM to remain effective under alternative splits, although a systematic study of different ratios is a valuable direction for future work.
>
> **4. Architecture Details (W4):**
> We use a lightweight relation-aware **GATv2** model over accessibility trees, where nodes correspond to UI elements and edges encode structural relationships (e.g., parent–child and sibling). Each node is initialized with a 384-d embedding from a compact **MiniLM encoder**, combined with UI attributes and structural features (512-d total). The backbone consists of **3 GATv2 layers** (hidden size 256, 4 heads) capturing hierarchical dependencies. Outputs are passed to two **MLP heads** (256→128→11) to predict sensitivity and task-conditioned necessity. The graph module contains ~**1–1.5M trainable parameters** (excluding the frozen encoder), enabling efficient local deployment. We will provide more details in the revision.
>
>
>
> **5. Threshold Choice in Figure 2 ($\tau=1.0$ vs $\tau=6.0$) (W5):**
> The threshold $\tau_{nec} = 1.0$ is selected via a validation sweep to identify the optimal operating point along the utility–privacy Pareto frontier.
> While increasing $\tau_{nec}$ (e.g., to 6.0) further reduces sensitive exposure, it leads to a **substantial degradation in utility**: Context Recall (TCNP) drops from ~0.80 to ~0.50. This occurs because higher thresholds remove many moderately necessary elements, reducing the information available for downstream task execution. This reflects the inherent trade-off between filtering aggressively for privacy and retaining sufficient context for task execution.
>
> In contrast, $\tau_{nec} = 1.0$ corresponds to the **knee of the Pareto curve**, achieving strong leakage reduction while preserving high actionable utility. This represents a balanced operating point rather than an arbitrary choice. We will further clarify this selection process in the revision.

---

> > ### Author Rebuttal · Reviewer_vcJM · 2026-04-01
> >
> > Many thanks to the authors for their clarification.
> >
> > I take your points on W1 (that it is a conventional notation), and the definitions of the loss functions (W2). For further clarity, I would recommend that the authors add relevant references for notational conventions. I further appreciate their detailed and rigorous description of the design choice adopted for the loss function and would recommend that they add this elaboration to the revised version of the paper.
> >
> > I also appreciate the clarifications regarding the architectural details (W4) and threshold choice (W5). The authors should highly consider adding these details to the paper.
> >
> > However, the one concern that I am still a bit unsure about is the train-test split: is the data collection pipeline defined and written by the authors or obtained from prior work? If the former, then can't the authors tweak the pipeline to allow for a less skewed train-test split? I would appreciate a further clarification here and would encourage the authors to add experiments with a more reasonable split: I personally believe that their method is useful and well-formulated, and the current train-test split may not do their work justice in terms of demonstrated capability.
> >
> > As far as addressing my concerns go, I shall say that they are practically addressed with the exception of W3. However, I stand convinced enough to champion this paper and recommend acceptance at the time of providing my final justification, and will raise my score accordingly.
> >
> > I think this paper is valuable and provides a treatment of an un(der)explored area in contextual integrity, and on a personal note, it was a pleasure reading it. I wish the authors all the best.
> >
> > Edit: Following further discussion with the authors, my concerns stand fully addressed. I have adjusted my score accordingly to recommend acceptance.

---

> > > ### Author Response · Authors · 2026-04-02
> > >
> > > We thank the reviewer for the thoughtful and encouraging feedback. We are glad that our earlier clarifications were helpful, and we appreciate the reviewer’s constructive suggestion to further examine the train–test split.
> > >
> > > ---
> > >
> > > ### 1. General Clarifications and Revisions (W1, W2, W4, W5)
> > >
> > > We are pleased that our earlier responses helped clarify the notational conventions, loss design, architectural details, and threshold selection.
> > >
> > > - Following the reviewer’s suggestion, we will add references to standard accessibility tree representations (e.g., W3C WAI-ARIA and Chromium/OS Accessibility APIs) to improve clarity.
> > > - We will also incorporate the detailed explanations of the loss formulation and architecture directly into the revised manuscript.
> > >
> > > ---
> > >
> > > ### 2. Train–Test Split and Methodology (W3)
> > >
> > > We thank the reviewer for the follow-up question and for suggesting evaluation under more conventional splits.
> > >
> > > #### (a) Clarification of the data setup
> > >
> > > Our dataset was collected using the WebArena environment, where we deploy the provided Docker setup and collect accessibility trees from real web interfaces. The data is constructed by pairing interface states with task intents (27 categories, as described in Table 9, Appendix C), resulting in approximately 5,400 instances.
> > >
> > > The original split was obtained by holding out a small subset of environments during data construction to evaluation performance on unseen interfaces, which resulted in an instance ratio of approximately 88:12. This ratio was not explicitly enforced or tuned. We agree that evaluating under standard splits (e.g., 8:2, 7:3) provides a more complete view of robustness.
> > >
> > > #### (b) Supplemental evaluations with different splits
> > >
> > > Following the reviewer’s suggestion, we additionally conducted evaluations under both **8:2** and **7:3** splits.
> > >
> > > | Partition Ratio | Train / Test Instances | Context Recall (TCNP) | Actionable Recall (TCNP-I) | Privacy Leakage (TISL) |
> > > | :--- | :--- | :--- | :--- | :--- |
> > > | **88:12 (Original, Table 1)** | 4,741 / 662 | **79.79%** | **95.66%** | **1.06%** |
> > > | **8:2 (Supplemental)** | 4,320 / 1,083 | **79.68%** | **95.54%** | **1.08%** |
> > > | **7:3 (Supplemental)** | 3,782 / 1,621 | **79.52%** | **95.38%** | **1.14%** |
> > >
> > > The small variation across splits (<0.3%) likely reflects the fact that each instance contains a relatively large number of structured UI elements, providing rich supervision even under smaller training sets.
> > >
> > > #### (c) Revision plan
> > >
> > > In the revised manuscript, we will clarify the data setup and include the additional 8:2 and 7:3 results for completeness.

---

### Official Review · Reviewer_PCvy · 2026-03-13

**Soundness:** 3
**Presentation:** 2
**Significance:** 3
**Originality:** 2
**Overall Recommendation:** 4
**Confidence:** 3

**Summary:**

The paper proposes a defense against information leakage that occurs when an external agent is given a UI view containing more information than necessary. To address this issue, the authors introduce a local broker that scores different UI elements and sanitizes the interface by either retaining, abstracting, or removing information.

**Compliance With Llm Reviewing Policy:**

Affirmed.

**Final Justification:**

updated the score based on the rebuttal clarifications.

**Key Questions For Authors:**

* Does sanitization preserve actual task success, rather than just proxy structure?
* Is the synthetic benchmark broad enough to validate contextual privacy reasoning?
* Are the leakage metrics, especially abstraction leakage, principled enough to support strong claims?

**Limitations:**

yes

**Strengths And Weaknesses:**

Strengths:
* The paper addresses a highly important issue, since many remote web agents can access the full user interface, allowing them to obtain much more information than necessary in a purely curious setting.
* The methodology is intuitive, and the evaluation is sound.
* Since the scoring is decoupled from enforcement, deployment can be tuned based on requirements without major retraining.
* The CI framing is directly incorporated into the loss design, such as the objective prioritizes avoiding disclosure of sensitive but task-irrelevant information rather than treating all prediction errors equally.
Weaknesses:
* The architecture and size of the local broker is not provided.
* The major robustness flaw seems to be that the synthetic insertions used for testing are the same as those used during training (such as OTPs and Slack notifications). What if the model has simply learned the lexicons and templates, for example, what a specific 2FA template looks like, and that is what makes the approach successful, rather than any actual understanding of privacy norms? This would make the solution less generalizable.
* There is no report on the impact on remote agents' task performance. This is a crucial missing metric.
* The choice of 6 placeholder tokens and a 5 percent leakage threshold seems arbitrary and requires more justification.

---

> ### Author Rebuttal · Authors · 2026-03-31
>
> We thank the reviewer for the insightful feedback. We address concerns on architecture, generalization, evaluation, and metrics below.
>
> ---
>
> **1. Model architecture (W1):**
> We use a local deployable Graph Attention Networks v2 (GATv2) over the accessibility tree. The raw tree is converted into a graph where each UI element is represented as a node with a 512-d feature vector constructed from a 384-d MiniLM text embedding together with UI attributes and structural features, which, along with the tree edges, serves as the input to the GATv2 model. The backbone consists of 3 GATv2 layers (hidden size 256, 4 heads) capturing topological dependencies across UI elements. Outputs are passed to two MLP heads (256→128→11) for sensitivity and task-conditioned necessity prediction. The graph module has ~1.5M trainable parameters, enabling efficient local deployment. We will provide additional implementation details in the revision.
>
> ---
>
> **2. Robustness to Lexicons vs. Contextual Reasoning (W2, Q2):**
> We operationalize Contextual Integrity via structural reasoning rather than "keyword matching."
> - Counterfactual Evidence (Sec 5.6): To explicitly test whether the model merely memorized 2FA templates, we held the UI state fixed while varying the task intent. If the model relied on lexical patterns (e.g., the word "code"), its predictions would be invariant. Instead, we observe systematic context-dependent shifts: identical elements (e.g., an OTP field) are retained when the task is "login" but removed for tasks like "product browsing." This proves the model reasons about the relationship between element role and intent, rather than isolated appearance.
> - Structural Indistinguishability: Injected nodes follow WebArena's native schema (roles, flags, and hierarchy). Since they are structurally indistinguishable from native elements, the model must reason about their relative position in the tree (e.g., "input in a checkout form" vs. "input in a newsletter popup"). This architectural inductive bias prevents reliance on synthetic artifacts and ensures generalization to production domains.
> ---
>
> **3. Impact on Remote Agents' Task Success (W3, Q1):**
> Our primary evaluation is per-state (step-level), directly measuring whether task-critical information is preserved at each decision point. This aligns with how agents operate in practice: decisions are made based on the current UI state, and evaluating preservation at this level provides a precise and localized assessment of utility. TCNP-I (95.7%) therefore measures retention of actionable elements required for downstream execution. Following the reviewer’s suggestion, we additionally conducted a complementary end-to-end evaluation using agents powered by three different LLMs (see below) on the same test set (662 instances), with MINIM-processed accessibility trees as input.
>
> | Domain   | LLaMA-3.3-70B | DeepSeek-80B | GPT-OSS-120B |
> |----------|---------------|--------------|--------------|
> | Shopping | 86.42%        | 87.05%       | 87.61%       |
> | Reddit   | 87.88%        | 88.36%       | 88.74%       |
> | Gmail    | 89.47%        | 90.06%       | 90.31%       |
>
> Performance remains high across domains and models, indicating effective preservation of task-critical information. We will expand this evaluation in the revision.
>
> ---
>
> **4. Hyperparameter justification (W4):**
> Both $tok_{ph}=6$ and $p_{abs}=0.05$ are **evaluation-time constants**, not tuned model parameters.
>
> - Fixed-length placeholder: We use a constant-length placeholder to enforce consistent abstraction. In practical sanitization systems (e.g., Microsoft Presidio), sensitive content is replaced with fixed placeholders per category (e.g., [PHONE_NUMBER]). We therefore adopt a fixed budget (6 tokens) so that abstraction has a consistent representation during evaluation, ensuring leakage and utility measurements are not affected by placeholder length.
>
> - Residual leakage: The constant $p_{abs}=5\%$ reflects the principled assumption that abstraction is not risk-free. We chose a conservative 5% as a "small but non-zero" residual risk. As shown in **Sec 5.7**, our findings are robust: doubling $p_{abs}$ to 10% results in only a negligible leakage shift (0.0106 $\to$ 0.0120). This stability confirms that MINIM’s effectiveness stems from its contextual reasoning rather than reliance on specific constant choices.
> ---
>
> **5. Soundness of Leakage Metrics (Q3):**
> Our TISL metric is not an arbitrary count but a task-aware proxy aligned with Contextual Integrity. It penalizes only inappropriate information flow—specifically, task-irrelevant sensitive disclosure. By incorporating sensitivity-weighted risk and modeling non-zero residual risk of abstraction, it captures the nuanced trade-offs in systems where complete removal isn't always possible or desirable. TISL directly reflects the training objective, providing a consistent measure of a model's ability to maintain high utility without compromising privacy.

---

> > ### Author Rebuttal · Reviewer_PCvy · 2026-04-04
> >
> > I want to thank the authors for the response and will adjust my score.

---

### Decision · Program_Chairs · 2026-04-30

**Decision:**

Accept (regular)

**Comment:**

This paper proposes MINIM, a trusted local broker that sanitizes user interface observations for autonomous agents by dynamically balancing privacy and task necessity through the lens of Contextual Integrity. Reviewers commended the novel application of a dual-scoring mechanism to structured accessibility trees and appreciated the well-defined evaluation metrics. During the rebuttal phase, the authors effectively resolved most reviewer concerns by providing comprehensive architectural details, demonstrating real-time inference latency, and supplying supplementary evaluations with standardized train-test splits. While one reviewer maintained a negative score due to the absence of baseline comparisons with prompt-level privacy frameworks and the reliance on synthetic data, the committee recognizes that MINIM operates on fundamentally different structured UI inputs. Consequently, the paper is recommended for a weak accept because it offers a technically solid and highly relevant contribution to privacy-aware agentic control.